# Study on the Release Law of Phenol during Water-Oil Shale Interaction Process

**Zhaoxia Sun, Qingyu Li, Quansheng Zhao and Shuya Hu ***

College of Environmental Science and Engineering, Qingdao University, Qingdao 266071, China;
szx19953532104@163.com (Z.S.); lqy910817@163.com (Q.L.); qszhaoqdu@163.com (Q.Z.)
* Correspondence: 90shuya@qdu.edu.cn

**Abstract:** Oil shale, as a reserve resource of conventional energy, has gradually attracted attention. However, water-rock interactions occur during in-situ shale oil extraction, and pollutants generated during this process can contaminate surrounding geological formations and groundwater environments. This article focuses on phenol produced by water-rock interactions and investigates the release behavior of phenol under different reaction temperatures and times, as well as how total organic carbon (TOC), total petroleum hydrocarbons (TPH), and pore size changes affect phenol. The study found that the release concentration of phenol increased with the increase in reaction temperature, reaction time, and the average pore size of the mineral. In addition, with the increase of TOC and TPH concentrations, the concentration of phenol also increased continuously.

**Keywords:** oil shale; water-rock interaction; phenol; release regularity

## 1. Introduction

With the increasing demand for conventional energy sources such as coal and oil, oil shale has attracted worldwide attention as an important substitute for conventional energy and a strategic reserve energy source. Compared with conventional energy reserves, oil shale resources have a huge storage capacity. The world's known oil shale resources can reach 40 trillion tons, and the shale oil produced can reach four times that of oil resources. Oil shale is a solid combustible organic mineral, light gray to brown in color, with a high ash (>40%) and organic matter content. It is deposited in sedimentary layers. Kerogen in oil shale is distributed in flattened strips with thicknesses from several micrometers to tens of micrometers [1]. Heating kerogen can produce oil and natural gas through pyrolysis [2–4], and its calorific value is generally >1000 Kcal/kg [5–7]. Although the composition of oil shale in different regions and countries varies, it mainly includes organic matter asphalt and polymer oil master, and its inorganic fraction is mainly composed of quartz, clay, carbonate, and silicates [8–12]. Oil shale has large reserves, wide distribution, high oil production rate, and great development potential, and because of its unique composition and structure, it can be applied to various fields such as energy, the chemical industry, building materials, and environmental protection [13], becoming a focus of global resource attention.

In-situ mining of oil shale refers to a method that directly acts on the underground oil shale layer, and heats the oil shale layer to cause the pyrolysis of the main organic matter of oil shale, producing oil and gas, and then extracting them to the surface through production wells [14]. Compared with surface retorting, in situ, mining has the advantages of a high conversion rate, less pollution, reduced land use and production costs, and will become the main way of oil shale mining [15]. Currently, the heating methods for oil shale in situ mining mainly include reactive heating, conductive heating, radiative heating, and convective heating. No matter which heating method is used, it will cause complex physical and chemical reactions and mineral transformations in oil shale [2,16] leading to the generation of a large number of pores and fractures in the oil shale layer. During the pyrolysis process of oil shale, the internal pore structure will change, these pores and

fractures are conducive to the migration and diffusion of oil and gas, which improves the yield and efficiency of in situ mining. At the same time, minerals in oil shale also undergo conversion. For example, fixed elements such as phosphorus, calcium, and organic matter will be released, which will affect the subsequent groundwater environment and ecological system [17]. Hu [18] conducted tests on the porosity of oil shale after heating at different temperatures and found that the porosity of oil shale increased continuously with the increase of heating temperature. The porosity of oil shale after heating at 500 °C could reach 19.1%, which is eight times that of unheated oil shale. The originally tight and impermeable, or weakly permeable, oil shale layer becomes a loose and weakly permeable layer, allowing groundwater to infiltrate into the pyrolysis zone [18,19]. Under natural conditions, the content of organic pollutants in groundwater is relatively low. However, when there is a hydraulic connection between oil-bearing strata and aquifers, the pollutants in groundwater will also undergo corresponding changes [20,21]. The pyrolysis products of oil shale are very complex, including hydrocarbon pollutants such as alkanes, alkenes, and polycyclic aromatic hydrocarbons, as well as non-hydrocarbon pollutants such as heavy metals, phenols, and benzene series compounds [22,23]. These pollutants can penetrate along fractures and pores, posing a potential risk of pollution to the groundwater environment. The degree of damage and consequences of this "stratum pollution source" to the groundwater environment is currently difficult to estimate.

From the Earth's surface to the deep interior, water-rock interactions are ubiquitous and ongoing. During the contact between groundwater and surrounding rock and soil, complex water-rock interactions constantly occur, changing the chemical composition of the surrounding rock and groundwater itself [24]. Based on this, some scholars have also considered that the water-rock interaction solution in oil shale layers may pose a potential risk of pollution to groundwater and have conducted a series of experimental studies. Amy [25] conducted a series of indoor continuous column leaching experiments to analyze the organic components leached from in-situ pyrolysis of oil shale, and identified that organic pollutants in the leachate would seriously deteriorate the groundwater quality within the mining area. Bern CR [26] studied the release characteristics of major elements, trace elements, and rare earth elements by immersing shale samples from multiple basins in the United States with different formation mechanisms in three different immersion liquids: brine, distilled water, and hydrochloric acid. Qiu [27] and He [28] conducted research on the pollution of groundwater by gas, oil, and residue generated by in-situ mining of oil shale. They immersed the pyrolyzed oil shale in water and detected sulfides, inorganic minerals, organic pollutants, heavy metals, and trace elements in the immersion liquid, and studied the migration and transformation pathways of these pollutants. Although many substances were detected, the research only focused on the study of pyrolyzed oil shale at room temperature and ignored the influence of temperature on the migration and release in water-rock interaction. Hu [17,29,30] conducted a series of water-oil shale and water-oil shale ash interaction experiments below 100 °C, and found that more TOC and TPH were released into the water along with the reaction time and temperature under low-temperature conditions. TOC stands for the total amount of organic carbon in both dissolved and suspended forms in an aqueous solution. It is usually used as a crucial indicator to assess the degree of organic pollution in water bodies [31,32]. TPH is a class of organic matter commonly found in organic contaminated sites, which is mainly composed of a mixture of alkanes, polycyclic aromatic hydrocarbons, and other organic substances and it is often used to indicate the degree of pollution of water bodies by petroleum hydrocarbon pollution. Meanwhile, not only does the in situ mining process of oil shale aggravate groundwater organic pollution, but continuous pollution of groundwater caused by the residual oil shale ash still exists. Overall, experimental research on water-oil shale interactions provides an important reference for an in-depth understanding of the impact of oil shale mining on the groundwater environment. It is just that most of the existing indoor immersion experiments have not taken into account the influence of temperature

factors, or have only studied water-rock interactions under low-temperature conditions due to experimental constraints.

Phenols are the main organic pollutants generated during the refining and processing of petroleum and coal mines. Phenolic pollutants are also produced during the exploitation of oil shale, including chlorophenol, nitrophenol, and other phenolic organic compounds [33]. Characterized by their high toxicity and non-biodegradability, they can persist in soils and waters for a long time, and have been blacklisted by the U.S. Environmental Protection Agency as a 129 priority controlled pollutant [34,35]. The phenolic compounds can cause chemical reactions between the cytoplasmic virus and the proteins in the cytoplasm of human cells, resulting in the loss of cell activity and causing serious harm to the human body. Currently, research on the interaction between water and oil shale mainly focuses on pollutants such as heavy metals, alkanes, and aromatic compounds, with less research on phenolic compounds [36,37]. Therefore, this research used oil shale samples from the Huadian district to conduct studies on water-rock interactions under multiple temperature gradients and reaction durations, to investigate the release law of phenols in water-oil shale interactions under high-temperature conditions after in-situ pyrolysis of oil shale.

The pore structure characterization analysis of oil shale samples under different reaction conditions was used to explore the influence of the pore structure of pyrolyzed oil shale on the release of phenols. This research can provide an important theoretical basis for environmental protection and management during water-rock interactions in the in-situ exploitation of oil shale.

## 2. Materials and Methods

### 2.1. Oil Shale Samples

Jilin Province is the region with the richest oil shale resources in China, accounting for 56% of China's proven oil shale resource reserves [38,39]. The paper selected Huadian oil shale as the research object (Figure 1).

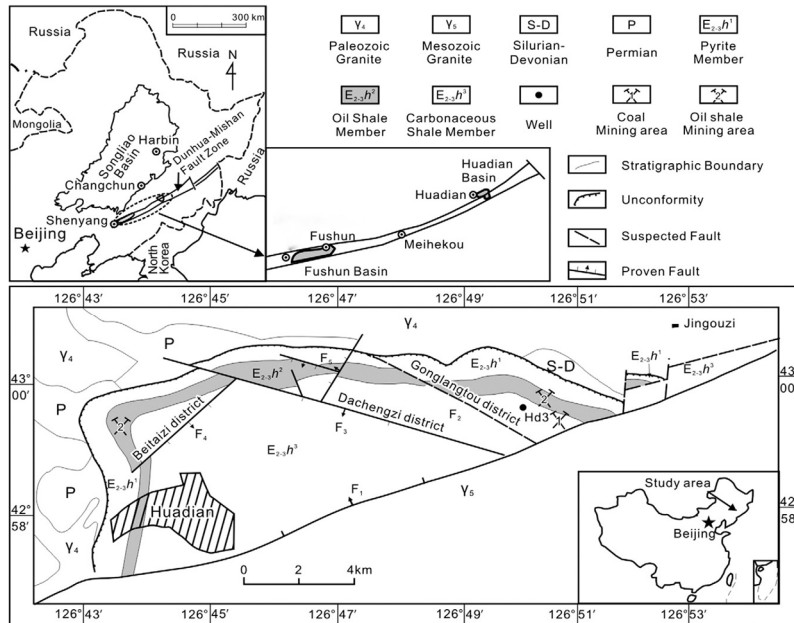

**Figure 1.** Geological map of Huadian basin, Jilin Province, NE China [40] (Sun, 2013).

Huadian Oil Shale Mining Area is located in Huadian City, Jilin Province, with an area of about 40 km$^2$, which is a low-mountain hilly area. The strata in the area have developed from old to new, including Permian, Jurassic, Cretaceous, Paleogene Neogene, and Quaternary strata. The Huadian oil shale sedimentary layer is located in the Paleogene Eocene Huadian Formation, the burial depth is shallow, generally 0–500 m. The number of

oil shale layers is large, and the general trend is from thin to thick to the north edge of the basin to the south with the sedimentary fault direction. The oil shale layer is sandwiched between the confined aquifer and the artesian aquifer, with a sampling depth of 110~120 m. The oil shale is grayish-brown, dense and blocky, with an oil yield ranging from 10% to 18%.

### 2.2. Experiment of Water-Rock Interaction

In order to explore the release mechanism of phenol during water-rock interactions under high-temperature conditions, water-rock interaction experiments were conducted under different reaction conditions. The oil shale was ground into powder and passed through a 30-mesh (0.6 mm) sieve, and then dried at 60 °C for 4 h. A mixture of 100 mL distilled water and 5 g oil shale powder in a ratio of 20:1 was then stirred and placed in a reaction vessel for reaction under the set temperature and duration.

Samples were put in a reactor, and the reaction temperature was set at 30 °C, 60 °C, 100 °C, 120 °C, 140 °C, 160 °C, 180 °C, 200 °C, 220 °C, 240 °C, 260 °C, 280 °C, respectively. There were 3 samples for each temperature bath. The samples need to be soaked for 1, 2, and 4 h, respectively. Since most scholars explore the reaction time of water-rock interaction, some 5 h, 72 h, a week, and some reach 30 days [2,21,41,42]. The temperature gradient is small and the content of phenol produced by the water-rock interaction in a short period of time is rarely involved, so the reaction time is set for a short time. After the reaction is completed, the aqueous solution samples were filtered and tested.

### 2.3. Detection Methods

#### 2.3.1. Phenol Detection Method

To determine phenol, we used the spectrophotometric method with 4-aminoantipyrine according to the standard of Water Quality—Determination of Volatile Phenols—Spectrophotometric Method with 4-Aminoantipyrine (HJ503-2009).

#### 2.3.2. TOC Detection Method

Before measurement, the reaction solution is filtered through a 0.22 μm membrane to ensure that there are no large particle suspensions or oil shale powder in the reaction solution. The organic carbon content in the reaction solution is determined using a Shimadzu SSM-5000A-TOC analyzer and a high-temperature catalytic oxidation method.

#### 2.3.3. TPH Detection Method

Determination of TPH concentration in the reaction solution according to the standard Determination of Water Quality petroleum ultraviolet spectrophotometry of China (HJ 970-2018).

#### 2.3.4. Microscopic Characterization Method of Pore Structure in Oil Shale

The changes in pore structure are observed using X-ray Diffraction (XRD) and Nitrogen Adsorption/Desorption methods [16,43,44]. XRD can be used to study the mineralogy of oil shale samples. This technique can provide information about the type and amount of minerals present in the rock matrix, which can affect the pore structure. Nitrogen adsorption/desorption is a widely used technique for characterizing the pore structure of porous materials. This technique involves measuring the amount of nitrogen gas adsorbed onto the surface of the oil shale sample as a function of pressure. The resulting data can be used to calculate the pore size distribution and specific surface area of the sample.

In this study, a fully automated multifunctional gas adsorption instrument (Micrometitics US ASAP 2020 Plus HD88) was used to perform low-temperature nitrogen adsorption experiments and LTNA characterization of Brunauer-Emmett-Teller (BET) specific surface area and Barret-Joyner-Halenda (BJH) pore structure parameters on seven oil shale samples, including raw samples and samples reacted at 180 °C, 200 °C, and 280 °C for 2 and 4 h in water-oil shale.

The first step was to obtain a representative sample of oil shale and crush it into small particles. The oil shale sample was loaded into the instrument of 3H-2000BET-A equipment and degassed at a low temperature to remove any remaining volatile components. The low-temperature nitrogen adsorption experiment was conducted with a heating rate of 10 °C/min, at a temperature of 30 °C, and a holding time of 10 min.

In this work, some of the prepared oil shale samples were ground further to suit XRD sample preparation requirements and subjected to XRD analysis using a Rigaku D/max-2200 X-ray diffractometer system with CuK$\alpha$ radiation ($\lambda$ = 1.5 Å). Scans were run from 0° to 90°, 2θ, with a step size of 0.05°/s. Quantitative identification of the minerals in the oil shale sample was achieved using the full pattern analysis technique known as the Rietveld method [16] using the Inorganic Crystal Structure Database (ICSD) [45].

## 3. Results and Discussion

### 3.1. XRD Analysis

According to the XRD test results (Figure 2), the inorganic mineral composition of oil shale in the Huadian area is mainly composed of quartz, kaolinite, orthoclase, and some iron minerals. Quartz accounts for 55.1%, followed by kaolinite (21.6%), orthoclase (11.3%), and montmorillonite (3%). XRD can detect the mineral composition in oil shale, but the main component of pollutant release in oil shale is kerogen, and water-rock interactions can be used to study the release of pollutants from kerogen.

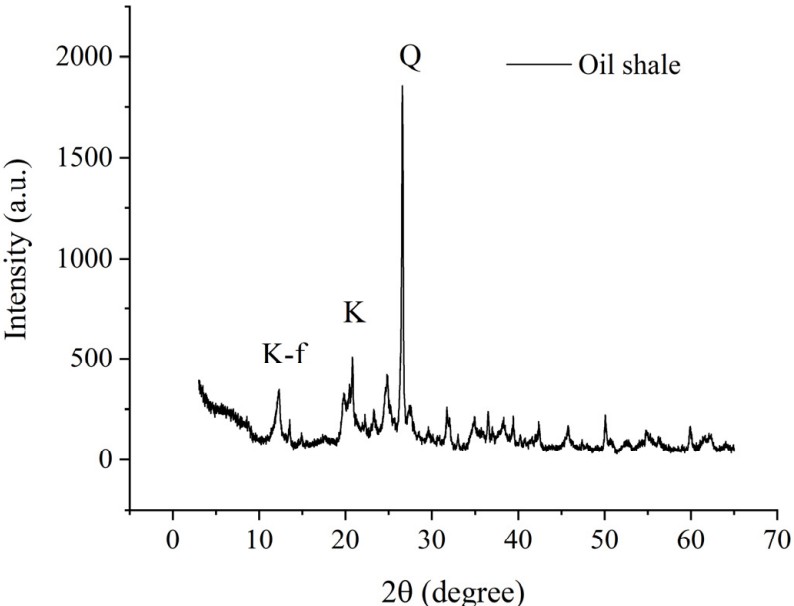

**Figure 2.** XRD analysis of oil shale in the Huadian area. Q—quartz; K—kaolinite; K-f—feldspar.

### 3.2. Phenol Content

The change in phenol content in the reaction solution is illustrated in Figure 3. Overall, the higher the reaction temperature and the longer the reaction time, the higher the phenol content in the solution. This can be attributed to two factors. Firstly, the pyrolysis of oil shale produces more phenol at higher temperatures. Secondly, phenol is only slightly soluble in water at room temperature, but it can be dissolved in water in any proportion at temperatures higher than 60 °C, and the amount of phenol dissolved increases with increasing temperature.

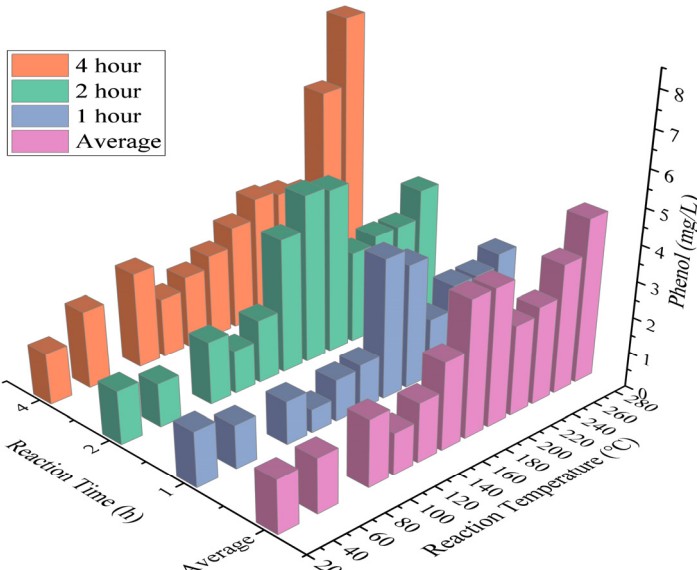

**Figure 3.** Changes in phenol content.

The solution obtained at 280 °C for 4 h showed the highest phenol concentration, which was 7.55 mg/L. The phenol content in the solution was relatively high when the reaction temperature exceeded 180 °C. The average phenol concentrations at reaction temperatures of 180 °C, 200 °C, 220 °C, 240 °C, 260 °C, and 280 °C were 3.85, 3.86, 2.54, 2.76, 3.63, and 4.61 mg/L, respectively. At reaction temperatures of 180 °C and 200 °C, the phenol content was higher than that at 220 °C under different reaction time conditions. After 200 °C, the phenol content showed a decreasing trend followed by an increasing trend with increasing reaction temperature.

When the reaction temperature was below 160 °C, the phenol content in the solution was relatively low, all of which were less than 2.5 mg/L. Although the average phenol concentration was the smallest (only 1.89 mg/L) at a reaction time of 1 h, it still far exceeded the requirements for phenol content in Class I and II water in the Chinese groundwater environmental quality standard (GB/T 14848-2017) (less than 0.001 mg/L). This clearly indicates that water-rock interaction during the pyrolysis process of in situ mining of oil shale has a significant impact on groundwater quality. The effects of temperature and reaction time on the volatilization of oil shale phenol are significant, and the impact on the groundwater environment cannot be neglected during the in-situ mining process with the infiltration of groundwater. In addition, changes in ionic strength and other organic matter produced by pyrolysis of oil shale Some other organic substances, such as benzene, toluene, etc., may interact with phenol or competitive adsorption, which may affect the content of phenol in aqueous solution. By analyzing the changes in TOC and TPH in the reaction solution, further exploration was conducted on the variation pattern of phenol content.

### 3.3. TOC Content

The trend of TOC in the reaction solution is shown in Figure 4. As the reaction temperature increase, the content of TOC in the reaction solution shows an upward trend. This can be attributed to two factors. Firstly, the pyrolysis of oil shale produces more organic matter at higher temperatures. Secondly, the water-rock interaction rate is increased with increasing reaction temperature and time [46]. However, when the reaction temperature exceeds 200 °C, the TOC content shows a decreasing trend followed by an increasing trend. Under the reaction condition of 180 °C, the TOC content in the reaction solution was the highest, with contents of 210.0 mg/L, 219.1 mg/L, and 238.5 mg/L after heating for 1 h, 2 h, and 4 h, respectively. Under the reaction condition of 200 °C, the TOC content in the solution decreased significantly, with an average content of only 138.85 mg/L. As the

reaction temperature increased, the TOC content gradually increased, with an average content of 161.1 mg/L under the reaction condition of 280 °C.

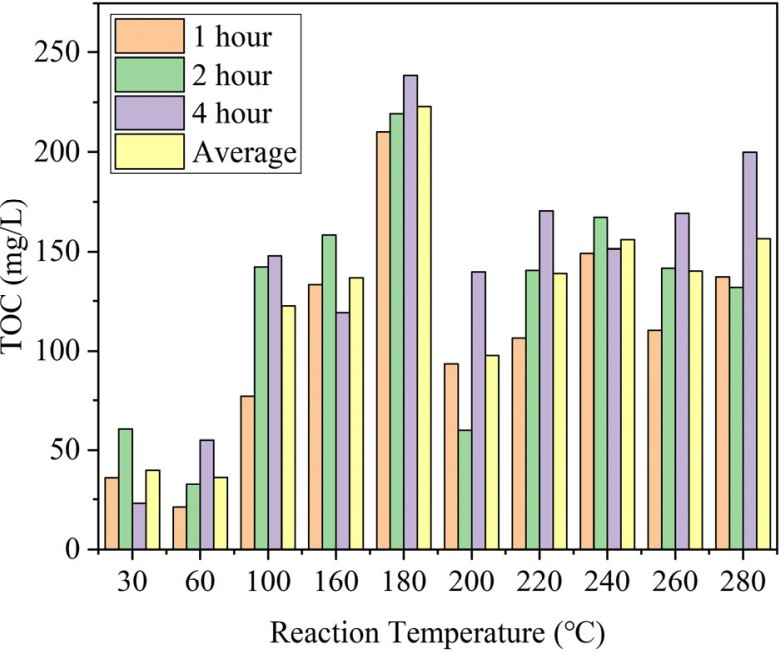

**Figure 4.** TOC release law.

Under the heating conditions of 180~200 °C, the pyrolysis of asphaltene in oil shale caused pore blockage, which affected the release of organic matter. At the same time, gas products were generated around 200 °C, and unstable carboxyl and ether groups were decomposed to form gases such as $H_2O$, $CO_2$, and CO [47], which also occupied a certain amount of pores, resulting in a decrease in the release of TOC. This is the reason why we speculate that the content of TOC decreases between 180 °C and 200 °C. As the reaction temperature continues to increase, the kerogen in oil shale is pyrolyzed, and the TOC content in the reaction solution increases accordingly.

*3.4. TPH Content*

Figure 5 shows the variation of TPH content in reaction solution at different reaction temperatures. The results indicate that TPH content in the solution increases with the rise of reaction temperature, and the highest TPH content is achieved in the reaction solution after 1 h at 180 °C, which reaches 6470.38 mg/L. Moreover, the duration of heating also affects the TPH content, and the TPH content after heating for 1 h is generally higher than that after 2 and 4 h at 180~220 °C. This may be due to the excessive volatilization of petroleum hydrocarbon compounds caused by heating for 4 h. The effect of temperature on TPH release is obvious, and high temperature accelerates the thermal decomposition of organic matter, causing the carbon-carbon or carbon–hydrogen bonds in petroleum hydrocarbons to break and release from oil shale. Under the reaction conditions of 180 °C, 200 °C, 220 °C, 240 °C, 260 °C, and 280 °C, the average TPH content is 5331.08 mg/L, 4809.90 mg/L, 5003.69 mg/L, 4582.72 mg/L, 4639.80 mg/L, and 4513.73 mg/L, respectively. After the reaction temperature exceeds 180 °C, the TPH content in the reaction solution begins to decrease, which may be caused by the volatilization of petroleum hydrocarbon compounds or the degradation of sulfate ion electron acceptors in the pyrite contained in oil shale [48].

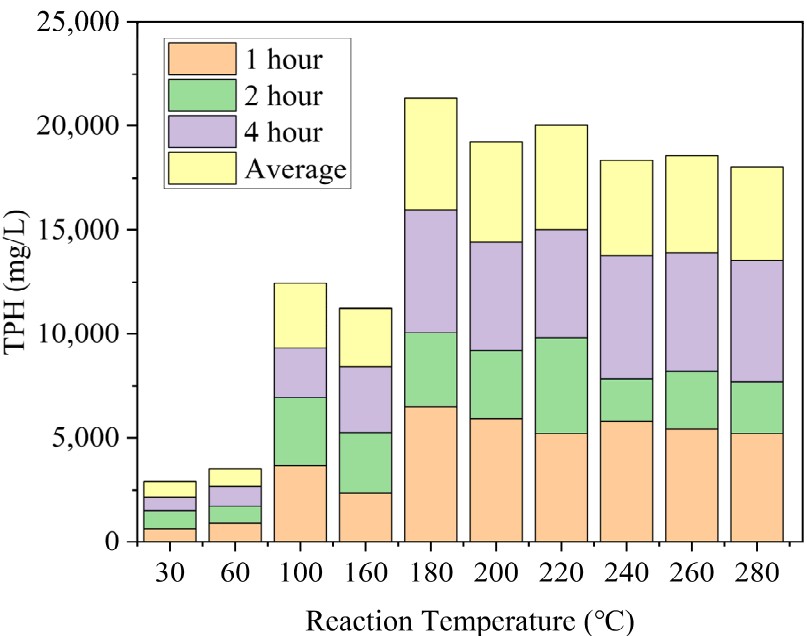

**Figure 5.** TPH release law.

### 3.5. Characteristics of Pore Structure in Oil Shale

Analysis of shale pore structure through low-temperature nitrogen adsorption can provide important parameters such as specific surface area, pore volume, average pore diameter, and adsorption capacity (Table 1), and observe the shape of the adsorption isotherm and the occurrence of hysteresis loops, thereby inferring the characteristics of shale pore structure. Figure 6 shows the adsorption-desorption isotherms of seven samples. According to the classification standard of adsorption isotherms by the International Union of Pure and Applied Chemistry (IUPAC), the adsorption isotherms of all seven samples belong to Type IV, which means that the adsorption capacity is small at low relative pressure and increases gradually and reaches saturation with the increase of relative pressure.

**Table 1.** Characteristics of pore structure parameters of oil shale.

| Number | Reaction Temperature (°C) | Time (h) | BET Surface Area (m$^2$/g) | BJH Total Pore Volume (cm$^3$/g) | Average Pore Size (nm) | Adsorption (cm$^3$/g) | Hole Pattern |
|---|---|---|---|---|---|---|---|
| O-0 | - | - | 11.700 | 0.029 | 8.200 | 18.763 | H3 |
| R180-2 | 180 | 2 | 10.388 | 0.018 | 6.840 | 14.761 | H3 |
| R180-4 | 180 | 4 | 11.374 | 0.021 | 7.933 | 17.346 | H3 |
| R200-2 | 200 | 2 | 11.293 | 0.021 | 7.659 | 17.570 | H3 |
| R200-4 | 200 | 4 | 12.585 | 0.026 | 8.880 | 23.413 | H3 |
| R280-2 | 280 | 2 | 9.774 | 0.021 | 8.613 | 18.757 | H3 |
| R280-4 | 280 | 4 | 9.684 | 0.024 | 10.224 | 21.704 | H3 |

In the region of relative pressure P/P0 < 0.4, the adsorption curve of shale rises slowly, indicating weak interaction between adsorbent molecules and shale pore surface, and only a small amount of adsorption sites is occupied. This stage is the transition stage from monolayer adsorption to multilayer adsorption, indicating the presence of micropores or mesopores with strong adsorption capacity in the sample. With the increase of relative pressure, the adsorption capacity increases, and when all active adsorption sites in the sample are occupied, the adsorption capacity reaches saturation.

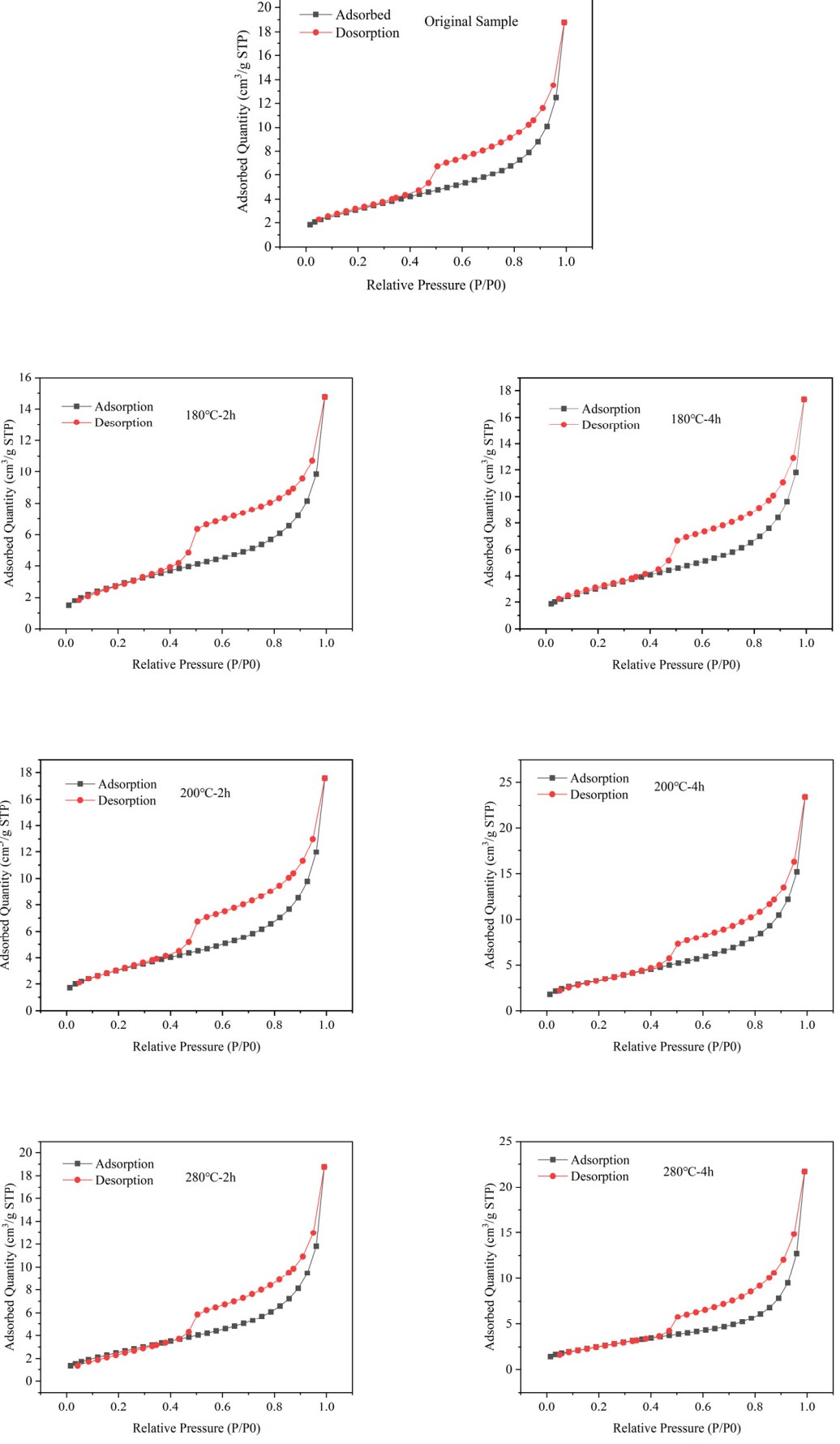

**Figure 6.** Isotherm diagram of low-temperature nitrogen adsorption-desorption.

At high pressure, the desorption branch and adsorption branch of all seven samples do not overlap at any temperature. In the high P/P0 region, capillary condensation occurs in the adsorbate, and the isotherm rises rapidly, and hysteresis can be observed. In the range of relative pressure 0.5~1, the desorption branch's adsorption capacity is significantly higher than that of the adsorption branch, and the adsorption isotherm exhibits a hysteresis loop. The type of hysteresis loop is closer to Type H3 (Hysteresis loop type by IUPAC), indicating the presence of slit-like structures, cracks, and wedge-shaped structures in shale pores. In addition, in the high-pressure stage (0.8~1), the adsorption curve does not show saturation, indicating that shale contains large or complex pores [49]. At the same time, at high pressure, some gases can still stay in the pores and not completely desorb, resulting in the adsorption and desorption loops not completely coinciding. This also indicates that the pore structure of shale is relatively complex, containing pores of various sizes and shapes, making the adsorption and desorption processes not fully coincide.

On the whole, the average pore size of the shale samples shows a relatively small variation range. Sample R280-4 has the largest average pore size of 10.224 nm, while sample R180-2 has the smallest average pore size of 6.84 nm. The saturated adsorption capacity of shale is closely related to its pore structure and pore size distribution. Shale with smaller pore sizes and larger pore volumes generally has a higher saturated adsorption capacity.

The sample R200-4 has the largest saturated adsorption capacity of 23.713 $cm^3$/g, followed by R280-4 with a capacity of 21.704 $cm^3$/g. The saturated adsorption capacities of R180-4, R200-2, and R280-2 are not significantly different from that of the original shale, all ranging from 17~18 $cm^3$/g. However, the saturated adsorption capacity of R180-2 is only 14.761 $cm^3$/g, a 21% reduction from the original shale sample. The trend of BJH Total Pore Volume change is consistent with the trends in average pore size and saturated adsorption capacity, with R280-4 and R200-4 samples higher than the original shale, while the BJH Total Pore Volume of other samples is generally lower than that of the original shale. In contrast to the trends in average pore size and saturated adsorption capacity, the specific surface area of the shale samples under the reaction conditions of 280 °C is the smallest, with R280-2 and R280-4 samples having specific surface areas of 9.774 $m^2$/g and 9.684 $m^2$/g, respectively. The specific surface area of the original shale is 11.7 $m^2$/g, and the specific surface areas of the other samples are not significantly different from that of the original shale [50–52].

Figure 7 displays the changes in Barrett–Joyner–Halenda average pore size under different reaction temperatures and heating times. Under the reaction condition of 180 °C, the average pore size of the oil shale sample was less than 8.0 nm, all smaller than the average pore size of the original oil shale sample. This trend is similar to that of Brunauer-Emmett-Teller specific surface area, which may be due to the blockage of some pores by the asphaltene produced during pyrolysis. Under the reaction condition of 200 °C, the average pore size of the sample after 2 h of reaction was smaller than that of the original sample, but the average pore size of the sample after 4 h of reaction was larger than that of the original sample. The average pore size of the oil shale samples after the 280 °C reaction was larger than that of the original samples, with the average pore sizes of the samples after 2 and 4 h of the reaction being 8.613 nm and 10.224 nm, respectively. We speculate that this is because the thermal stress on the oil shale exceeds the degree of particle cementation inside the oil shale, resulting in thermal cracking [53], which leads to an increase in the original small pore size and an increase in pore volume. High temperature has a greater impact on pore size changes, and an increase in reaction time also has a certain effect on pore size changes under lower temperature conditions.

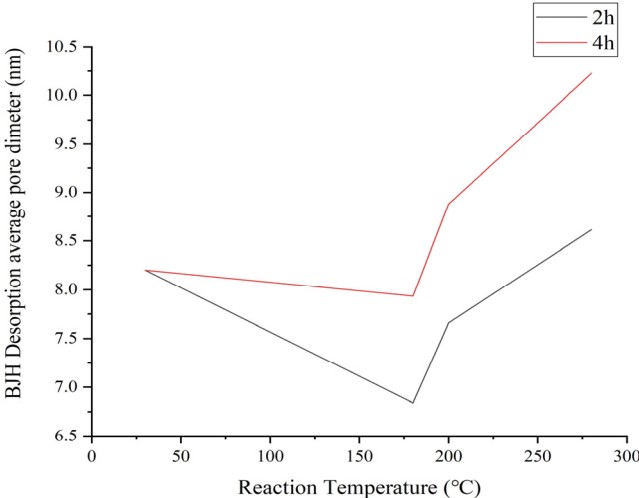

**Figure 7.** Reaction temperature and heating duration vs. average pore size of BJH.

At 180 °C, the concentrations of TOC and TPH in the reaction solution reached their maximum values, and the content of phenol also reached a high value at a reaction time of 1 or 2 h and a reaction temperature of 180 °C. However, the maximum value of phenol appeared in the 280 °C-4 h time period, during which the oil shale's average pore size also reached its maximum value of 10.224 nm.

## 4. Conclusions

Under the reaction condition of oil shale to water ratio of 1:20, the phenol content in the reaction solution significantly exceeded the maximum allowable levels for Class I and II groundwater as specified in the Chinese Groundwater Environmental Quality Standard (GB/T 14848-2017), which is less than 0.001 mg/L. This finding indicates that the production of phenol during in-situ pyrolysis has a potential impact on the quality of groundwater. Notably, the highest concentration of phenol, 7.55 mg/L, was observed in the solution obtained at 280 °C for 4 h. Furthermore, the phenol content was relatively high when the reaction temperature exceeded 180 °C. After 200 °C, the phenol content showed an initial decrease followed by an increase with increasing reaction temperature. Overall, both the TOC and TPH content demonstrated an increasing trend with increasing reaction temperature and reaction time.

The pore structure of reactive oil shale is complex, mainly consisting of micropores or mesopores. The range of variations in the average pore size of oil shale samples is relatively small under different reaction conditions. The trend of BJH total pore volume change is consistent with that of average pore size and saturated adsorption capacity, with most samples having a BJH total pore volume lower than that of the original oil shale. The asphaltene produced during the low-temperature pyrolysis process can block some pores. Under the reaction condition of 180 °C, the average pore size of the oil shale sample is less than 8.0 nm. Under high-temperature conditions, when the thermal stress on the oil shale exceeds the degree of particle cementation inside the oil shale, it leads to an increase in pore size and pore volume.

**Author Contributions:** Conceptualization, S.H.; writing and editing, Z.S.; methodology, Z.S.; analysis, Q.L. and Z.S.; supervision, Q.L.; Data curation, Q.Z.; project administration, S.H. All authors have read and agreed to the published version of the manuscript.

**Funding:** This research was funded by the National Natural Science Foundation "Study on the mechanism of BTEX release and aquifer pollution during in-situ exploitation of oil shale" of China (Grant No. 42002260).

**Data Availability Statement:** Not applicable.

**Acknowledgments:** The author acknowledges the comments of the reviewers.

**Conflicts of Interest:** The authors declare no conflict of interest.

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
