# Peer review of "Study on the Release Law of Phenol during Water-Oil Shale Interaction Process"

_water, doi:10.3390/w15112017_

Round 1

Reviewer 1 Report

Dear Authors,

 I was asked to review your work entitled “Study on the release law of phenol during water-oil shale interaction process”, you proposed for publication to Water.

I am not strictly an expert on this topic; anyway, I find your research offers interesting data on the relations among  temperature, formation of phenols, and porosity. Thus, the manuscript could be relevant to a wide audience, and it may be considered for publication, after some minor adjustments.

Accordingly, I will suggest the Editor to accept your manuscript after minor revisions.

You can find my line-by-line comments in the attached file; here, some points I would like to remark:

- I am not a native English, thus I usually avoid to propose changes of language and style. Nonetheless, please consider a general review of English (not fully clear sentences, use of verbs, etc.).

- Please check the style adopted for citations by the journal, for both in-text citations and reference list.

- Abstract: the second part (lines 12-20) contains too much detail and could be simplified.

- Introduction: this section broadly considers the “state of the art”, although I am not able to say if all the main sources have been cited. If possible, add some more detailed geographic  and geological information about the sampled unit (outcrop area, summarized geological context, lithological and sedimentological features, localization of oil, etc.): this information is needed to evaluate the applicability of your results beyond the case study.

- Methods: some information should be moved to the introduction (e.g., lines 165-167, 173-177, 181-183), to the results (e.g., lines 209-214), or removed at all (e.g., lines 148-152, 159-161).

- Results and discussion: usually, these two sections are separated, but it is your choice. Nonetheless, please be careful with generalizations of the results of the case study (e.g., lines 238-242). Also please consider (in the Conclusions section as well) you tested porosity rather than permeability, and more information is needed to speculate about flows of pollutants to groundwater or soils.

Hope my comments could have been of some utility at this stage.

Kind Regards

The reviewer

Author Response

Dear Professor,

We greatly acknowledge your constructive comments and advice. Thank you for your patience and these valuable comments. We attach great importance to each of your comments. We have carefully revised our manuscript according to your suggestions. Our changes were detailing the attachment.

Reviewer 2 Report

The paper submitted for review refers to the study on the release law of phenol during water-oil shale interaction process. This research is very important, especially that the phenol content in water is a standardized parameter and very unfavorable to health.
However, I have a few remarks:
1. The introduction is extremely extensive and provides an excellent background for the conducted research, however, I recommend referring to more recent literature.
2. Please provide details in chapter 2.2. Experiment of Water-rock Interaction, why such time intervals were used for the indicated temperatures. Was the experiment also carried out for longer periods?
3. I also lack information on the impact of phenol on the environment as well as national and global legal standards (what is included in the conclusion is a bit lacking).
In addition, the article is a very important contribution to the research information base. It is well written, the research is thoroughly presented, the figures are legible.
The publication is a valuable source of information and forms the basis for further legal analyzes and implementation of new solutions.

Thank you for considering my opinion. I encourage the authors to continue working on improving the manuscript.

Author Response

Dear Professor,

We greatly acknowledge  for your constructive comments. Thank you for your patience and these valuable comments. We attach great importance to each of your comments. We have carefully revised our manuscript according to your suggestions.  The following attachment is our reply to the comments.
